# Antidiabetic Effects of *Pediococcus acidilactici* pA1c on HFD-Induced Mice

**DOI:** 10.3390/nu14030692

**Published:** 2022-02-07

**Authors:** Miriam Cabello-Olmo, María Oneca, María José Pajares, Maddalen Jiménez, Josune Ayo, Ignacio J. Encío, Miguel Barajas, Miriam Araña

**Affiliations:** 1Biochemistry Area, Department of Health Science, Public University of Navarre, 31008 Pamplona, Spain; miriamcabelloolmo@gmail.com (M.C.-O.); mjose.pajares@unavarra.es (M.J.P.); ignacio.encio@unavarra.es (I.J.E.); 2Genbioma Aplicaciones SL., 31191 Esquíroz, Spain; maria@genbioma.com (M.O.); josune@genbioma.com (J.A.); 3IDISNA Navarra’s Health Research Institute, 31008 Pamplona, Spain; 4Division of Hematological-Oncology, CIMA, University of Navarre, 31006 Pamplona, Spain; mjimenezan@unav.es

**Keywords:** *Pediococcus acidilactici*, fasting blood glucose, insulin sensitivity, prediabetes, diabetes, high-fat diet, probiotics, gut microbiota and dysbiosis

## Abstract

Prediabetes (PreD), which is associated with impaired glucose tolerance and fasting blood glucose, is a potential risk factor for type 2 diabetes mellitus (T2D). Growing evidence suggests the role of the gastrointestinal microbiota in both PreD and T2D, which opens the possibility for a novel nutritional approach, based on probiotics, for improving glucose regulation and delaying disease progression of PreD to T2D. In this light, the present study aimed to assess the antidiabetic properties of *Pediococcus acidilactici* (pA1c) in a murine model of high-fat diet (HFD)-induced T2D. For that purpose, C57BL/6 mice were given HFD enriched with either probiotic (1 × 10^10^ CFU/day) or placebo for 12 weeks. We determined body weight, fasting blood glucose, glucose tolerance, HOMA-IR and HOMA-β index, C-peptide, GLP-1, leptin, and lipid profile. We also measured hepatic gene expression (G6P, PEPCK, GCK, IL-1β, and IL-6) and examined pancreatic and intestinal histology (% of GLP-1^+^ cells, % of goblet cells and villus length). We found that pA1c supplementation significantly attenuated body weight gain, mitigated glucose dysregulation by reducing fasting blood glucose levels, glucose tolerance test, leptin levels, and insulin resistance, increased C-peptide and GLP-1 levels, enhanced pancreatic function, and improved intestinal histology. These findings indicate that pA1c improved HFD-induced T2D derived insulin resistance and intestinal histology, as well as protected from body weight increase. Together, our study proposes that pA1c may be a promising new dietary management strategy to improve metabolic disorders in PreD and T2D.

## 1. Introduction

Prediabetes (PreD) refers to an intermediary phase between healthy state and diabetes disease where glycaemia is altered but still below the threshold for the diagnosis of diabetes. PreD is characterized by an impaired glucose tolerance (IGT) and fasting blood glucose (FBG) [1], and it is associated with insulin resistance (IR), beta-cell dysfunction, excessive body fat, inflammation, and dyslipemia [2,3]. In fact, 5–10% of people with prediabetes progress to T2D every year, over 25% within 3–5 years, and 70% during their lifetime. On top of this, PreD can, itself, provoke severe complications in tissues and organs, increase microvascular and macrovascular disease, and act as a potential risk factor for type 2 diabetes (T2D) and cardiovascular diseases [1,3,4]. This aspect is extremely relevant since T2D involves serious complications that affect the quality of life and healthcare costs [5]. PreD occurrence has worryingly raised in past years, and current predictions estimate a greater global prevalence for the coming decades to the extent that it will reach epidemic proportions worldwide [1]. The prevalence of prediabetes ranged from 9.6% to 37.2% worldwide. PreD is a reversible metabolic condition by changes of lifestyle (diet, exercise, and weight loss). However, current evidence indicates that diabetes prevention programs, based only on lifestyle change, have not been successful in preventing T2D in people with isolated impaired fasting glucose.

At this juncture, mounting evidence has suggested that the gastrointestinal microbiota (GM) is a relevant environmental factor in PreD and T2D development. The proposed mechanistic links include changes in intestinal permeability, energy homeostasis, metabolism, and especially inflammation, which is closely related to IR [6,7,8,9]. The literature demonstrates that diabetic subjects present distinct microbiota. To illustrate, it has been described that GM in diabetic patients is characterized by lower gene count (abundance), a decline of butyrate-producing bacteria, and an enrichment of opportunistic microorganisms [10,11]. Several lines of evidence suggest that GM may also play a role in T2D development and progression. However, the causality between gut dysbiosis and T2D remains enigmatic. A recent study showed that subjects that went on to develop T2D already presented a distinct GM composition prior to the dysregulation in glucose homeostasis [12]. Besides, another study reported that GM composition changes during the course of the T2D, since subjects with PreD presented microbiota features different from those observed in healthy individuals or those with T2D [13]. On top of that, GM also mediates the effect of antidiabetic drugs [9,14,15], and findings from human studies indicate that individuals under pharmacological treatment presented a different GM composition [16]. In this scenario, the GM appears as an exciting new therapeutic target for PreD and T2D.

Dietary factors are important microbiota modulators [17] and, at the same time, contributing factors for PreD and T2D development [5]. Thus, a number of dietary strategies involving fermented foods [18,19], functional foods [20], postbiotics [21], and probiotics [22,23,24] are currently being investigated for their effectiveness in addressing diabetes-associated dysbiosis and preventing, or reversing, diabetic complications. Probiotics can alter GM’s composition and activity, improve the gut mucosa, modulate the immune system, and protect against pathogens [25,26], and for those reasons, they have been investigated in several diseases and conditions [27,28]. Concerning T2D, animal and human studies have demonstrated that probiotics can improve glycemic control, reduce IR and inflammation, restore mucus production, and reinforce the intestinal barrier function [29,30,31]. The vast majority of these investigations are focused on *Lactobacillus* or *Bifidobacterium* spp. [32,33]. However, in the last years, other bacteria taxa have gained importance for targeting T2D [30,31]. 

The present research revolves around a newly identified *Pediococcus acidilactici* strain, *Pediococcus acidilactici* CECT 9879 (pA1c), a Gram-positive lactic acid-producing bacteria, Previous experimental works on other PA strains showed good probiotic properties, including tolerance to acid and bile salts, and resistance to gastrointestinal simulated conditions [34,35,36,37], good aggregation, coaggregation and adhesion properties [34,38], antagonistic activity against pathogens [35,38,39], and production of bacteriocins [40]. In addition, animal studies have indicated beneficial effects on models of constipation [41], infection with *Clostridium difficile* [42], colitis [43], and dermatitis [44]. However, to our knowledge, its antidiabetic properties have not yet been investigated. Following this target, this study deals with the in vivo antidiabetic activity of a *Pediococcus acidilactici strain* (pA1c) in diabetic mice, induced by high fat diet (HFD). Here, we hypothesize that this probiotic could improve glycemic control. In such case, it would be regarded as a helpful, nutritional approach for preventing or delaying progression to T2D in at-risk population.

## 2. Materials and Methods

### 2.1. Experimental Design

Twenty-four male C57BL/6 mice (Charles River Laboratories), aged 9 weeks, were acclimated for 2 weeks. The animals were housed in groups of 6 animals in a controlled environment (a room with constant temperature and humidity under a 12:12 h light-dark cycle). The animals were randomly divided and allocated into two groups (*n* = 12 each) [1] Control group (C group), animals receiving diet plus placebo; [2] Treated group (T group), animals receiving diet plus a probiotic formulation with pA1c. This strain was deposited, according to the Budapest Treaty, in the Spanish Collection of Type Cultures (CECT) with identification reference CECT 9879 from the proprietary strain collection of Genbioma Aplicaciones S.L. backed by an international patent [PCT/EP2020/087284]. This species meets the criteria of qualified presumption of safety (QPS) by the *European Food Safety Authority* (EFSA) [45]. Both placebo and probiotic formulation were provided by Genbioma Aplicaciones S.L. (Navarra, Spain).

Following the acclimation period (T0), animals continued with a normal standard diet (ND) enriched with either placebo or probiotic formulation during 2 weeks. From that time (T2) and thereafter, animals received a HFD with 60% kcal from fats (TD.06414, Envigo, Indianapolis, IN, USA; Tekla, Kennesaw, GA, USA), as we have already described in previous publications [46].

The animals had *ad libitum* access to the diets and water, and food acceptance and consumption was visually confirmed by one investigator. Body weight (BW) and FBG were checked weekly. Animals were sacrificed at 12 weeks, by cervical dislocation, for the collection of tissues and blood samples. Liver tissues were stored at −80 °C, and pancreas and small intestine tissues were fixed in 10% formaldehyde solution during 24 h. Blood samples were centrifuged for 8 min at 2000 rpm, and serum samples were stored at −80 °C. The experimental design is available in Figure 1. All animal procedures were performed under protocols approved by the Institutional Committee on Care and Use of Laboratory Animals (CEEA, University of Navarra) (Protocol number: CEEA/017-20). 

### 2.2. Experimental Diets

Commercial diets (ND and HFD) were enriched with either the placebo or the probiotic formulation every week. The amount of probiotic formulation was adjusted at a daily pA1c dose of 1 × 10^10^ CFU per animal. The LAB counts in the diet + probiotic formulation were performed by plate counting on MRS agar (Scharlau, Barcelona, Spain), incubated for 48 h at 37 °C under CO_2_ atmosphere (5%).

### 2.3. Body Weight, Fasting Blood Glucose and Glucose Tolerance Test 

Body weight was measured once a week with an electronic balance. 

Fasting (24 h) blood glucose determination was assayed weekly. Blood samples were obtained from the tip of the tail vein, and glycemia was measured using a glucometer (Accu-chek Aviva, Roche, Basel, Switzerland). 

Glucose tolerance was assessed after 2, 4, and 8 weeks of treatment. Glucose tolerance tests (GTTs) were determined in 24 h fasted animals at different time points (baseline, 20, 40, 60, and 120 min) after glucose (Baxter, Valencia, Spain) intraperitoneal injection (2 mg/kg of body weight). The area under the curve (AUC) of glucose values was assessed for each group from 0 to 120 min. 

### 2.4. Serum Biochemical Analysis

In order to determine the lipid profile, triglycerides (TG), free fatty acids (FFA), and total cholesterol (TC) serum concentrations were measured with colorimetric assays using commercial kits (Sigma-Aldrich, St. Louis, MO, USA).

### 2.5. C-Peptide, GLP-1 and Leptin Analysis 

Serum C-peptide, GLP-1, and Leptin concentrations were quantified with commercial Enzyme-linked immunosorbent assay (ELISA) kits (Abyntek Biopharma, Derio, Spain).

Both HOMA-IR and HOMA-β were determined at the end of the study. HOMA-IR was calculated by the formula: HOMA-IR = serum C-peptide (ng mL^−1^) × blood glucose (mmol L^−1^)/22.5. HOMA-β was calculated by the formula: HOMA-β = 20*serum C-peptide (ng mL^−1^)/(blood glucose (mmol L^−1^) − 0.35).

### 2.6. Tissue Collection and Histological Analysis

Fixed tissue samples (pancreas and small intestine) were embedded in paraffin, cut into 3 µm thick sections, and stained with hematoxylin and eosin (H&E) for light microscopy examination (*n* = 6 for each group). An additional section of small intestine was also stained with periodic acid-schiff (PAS). The slides were digitized using a histology slide scanner Aperio CS, running under the Scan Scope Console software (v.10.2.0.2352, Leica Biosystems, Inc. 1700 Leider Lane Buffalo Grove, IL, USA).

The villus length was analyzed in H&E-stained sections from the small intestine. Five longest villi per slide were measured in twelve different sections of each animal. In PAS-stained small intestine sections, the percentage of goblet cells per total enterocytes was calculated in ten random areas per animal. 

For immunohistochemical analysis, sections were dewaxed in xylene and rehydrated through a graded alcohol series. Endogenous peroxidase activity was blocked by placing sections in 3% hydrogen peroxidase for 10 min. For GLP-1 immunolabelling, microwave antigen retrieval was carried out with EDTA buffer (1 mmol/L, pH 8) for 30 min. Nonspecific binding sites were blocked with 5% goat normal serum in TBS-Tween (Wash buffer, Dako, Glostrup, Denmark) for 30 min. Subsequently, tissue sections were incubated with the primary antibody (pancreas with an anti-insulin dilution 1:8000; Dako, A0564 and small intestine with an anti-GLP-1 dilution 1:2000; Abcam, ab26278) overnight at 4 °C. After a rinse with TBS, incubation with a HRP secondary antibody (dilution 1:100, P0141 Dako) or Envision complex (Dako) was carried out. Finally, immunostaining was developed with diaminobenzidine, lightly counterstained with hematoxylin, and mounted in distyrene plasticizer xylene. Tissues expressing different levels of antigen were included in each immunohistochemical run to control variation between experiments. Positive and negative controls were included in each experiment. Negative controls were done, leaving out the primary antibody. As positive control, we used tissue previously shown to express the antigen of interest. 

For insulin analysis, positive insulin areas and total area were measured in five representative photographs taken at 10× magnification from four different sections per animal. In addition, with the purpose to study the intensity of the insulin-positive areas, the staining intensity was scored using the following criteria: *1, weak intensity; 2, moderate intensity; 3, strong intensity*. Subsequently, the percentage of weak, moderate, and strong intensity among the total insulin-positive areas was determined for each animal and study group. For GLP-1 analysis, GLP-1 positive cells, total cells, positive GLP-1 area, and total area were measured in ten random fields (magnification 20×) from eight different sections per animal. Quantification analysis for insulin and GLP-1 was performed using Image J 1.53 software (U.S. National Institutes of Health, Bethesda, MD, USA).

### 2.7. Quantitative Real-Time PCR (RT-qPCR) 

Frozen (−80 °C) liver tissues were used for total RNA extraction, using RNeasy Mini Kit (Qiagen, Germantown, MD, USA) according to the manufacturer’s instructions. The purity and concentration was determined by spectrophotometry using Nanodrop One (Fisher Scientific S.L.. C/Luis 9 28031 Madrid. Spain). The samples with total RNA were treated with DNase I (amplification grade, Sigma-Aldrich) to eliminate any DNA molecule. For that, 1 µg of RNA was treated with 1 µL of 10× DNAse I Reaction buffer, 1 µL of DNase I, and 3 µL of PCR grade water. Following a 15 min incubation at room temperature, 1 µL of 25 Mm EDTA was added to stop the reaction. The tubes were then incubated at 65 °C for 10 min and placed on ice. We used 10 µL of treated RNA to synthesize cDNA templates using a reaction mixture containing deoxynucleotides (dNTPs) mix, random primers, M-MLV reverse transcriptase, RNaseOUT recombinant ribonuclease inhibitor, and PCR grade water, following the manufacturer’s protocols. All reagents were purchased from Invitrogen, except for the water and dNTPs, which were purchased from ThermoScientific.

The mRNA expression levels of hepatic glycogenesis related genes (glucose-6-phosphatase (G6pase), phosphoenolpyruvate carboxykinase (PEPCK), as well as glucokinase (GCK)), inflammation-related genes (interleukine-1β (IL-1β), and interleukine-6 (IL-6)) were analyzed and further normalized using glyceraldehyde-3-phosphate dehydrogenase (GADPH) as a house-keeping gene. Reverse transcription-generated cDNAs were amplified in a 20 µL reaction using RT-qPCR in a CFX Connect TM Real-time system (Bio-Rad, Hercules, CA, USA) with iQ™ SYBR^®^ Green Supermix (Bio-Rad, Hercules, CA, USA). The reaction buffer consisted of 2 µL of the synthesized cDNA, 0.4 µL of F and R primers, 10 µL of iQ™ SYBR^®^ Green Supermix (2×), and 7.2 µL of PCR grade water. Samples were run in triplicates for each target gene. Each PCR program included a first denaturation step at 95 °C for 3 min, followed by 40 cycles of denaturation at 95 °C for 10 s, annealing for 30 s, and extension at 62 °C for 30 s. Melting curve analysis was performed to confirm amplicon specificity by slow heating from 65 to 95 °C, in increments of 0.5 °C s^−1^, with continuous fluorescence collection. Data is expressed as the relative mRNA normalized to GADPH and analyzed according to the comparative cycle threshold method (2^−∆∆CT^). Primer sequence for the targeted mouse genes and sources are shown in Appendix A.

### 2.8. Statistical Analysis

All statistical procedures were performed using SPSS software for Microsoft (IBM SPSS Statistics 27, Armonk, NY, USA). Data were submitted to Student’s t-test or Mann–Whitney U test as appropriate. The significance level was set to *p* < 0.05, and *p* < 0.01 and *p* < 0.001 were considered highly significant and extremely significant, respectively. Data are presented as mean ± standard deviation (SD). Pearson’s correlation coefficients were examined to determine any relationship between parameters. Data from FBG underwent logarithmic transformation before calculations.

## 3. Results

### 3.1. pA1c Supplementation Attenuates Body Weight Gain 

No statistically significant differences were found in the BW at the beginning of the study between both groups (24.4 ± 2.1 versus 23.6 ± 1.9 g C and T groups, respectively; *p* > 0.05). From the second week and onwards, coinciding with the HFD administration, the mean values of T group separated from C group values, and such differences were statistically significant (*p* < 0.05) at 4, 5, 6, 9, 11, and 12 weeks. The BW differences at the endpoint were 39.6 ± 4.6 g and 34.2 ± 2.3 g, in C and T groups, respectively (*p* < 0.05) (Figure 2A). As expected, both groups presented a rise in BW compared to their baseline values and such a rise was greater in C group (62.3% BW gain) than T group (44.8% BW gain) (Figure 2B).

### 3.2. pA1c Improves Fasting Blood Glucose Levels and Glucose Tolerance Test

pA1c administration exerted a glucose-regulating effect after 12 weeks of administration (Figure 3A,B). At the beginning of the study, both groups presented comparable FBG values (3.8 ± 0.5 versus 3.8 ± 0.4 mmol/L in C and T groups, respectively; *p* > 0.05). At 1 week, T group presented greater FBG values compared with those in C group (*p* < 0.05). Nevertheless, from 3 weeks onwards, this trend was inverted, and T group presented statistically significant lower FBG values compared with those in C group (*p* < 0.001). At the end of the study, FBG was up to 31% lower in T group (7.8 ± 0.4 versus 5.9 ± 0.5 mmol/L in C and T groups, respectively; *p* < 0.001). 

As expected, since animals received a HFD, they presented a rise in FBG values during the course of the study. Such a rise, however, was marked more in C group (107.9 versus 56.5% rise in FBG in C and T groups, respectively).

After 2 and 4 weeks of supplementation with pA1c, no statistical significant differences were found in GTT. However, after 8 weeks of supplementation, values from GTT indicate notable differences between experimental groups, which reached statistical significance at 0 (*p* < 0.01), 20 (*p* < 0.01), and 40 min (*p* < 0.01) (Figure 3C). Glucose AUC value in T group was up to 18.8% lower than in C group (1758.7 ± 271.6 and 1428.2 ± 283.0 in C and T groups, respectively; *p* < 0.01) (Figure 3D). All these data suggest that the exposition to pA1c in mice fed with HFD improved glycemic control.

### 3.3. pA1c Promotes Insulin Secretion and Preserves β-pancreatic Islets Function

Insulin secretion, measured as serum C-peptide levels, was statistically significantly higher in T group (306.8 ± 42.0 and 356.3 ± 56.6 pg/mL C and T group, respectively; *p* < 0.05) (Figure 4A). When HOMA-IR and HOMA-β indices were calculated, we found that treated animals presented a greater insulin sensibility (HOMA-IR: 0.11 ± 0.01 and 0.09 ± 0.01 in C and T groups, respectively; *p* < 0.01) (Figure 4B), as well as better β-cell functionality (HOMA-β: 0.81 ± 0.11 and 1.31 ± 0.21 in C and T groups, respectively; *p* < 0.001) (Figure 4C).

Pancreas histopathological examination revealed that the ratio of insulin-positive area and total area was comparable between both experimental groups (3.7 ± 0.4 and 2.9 ± 1.0% in C and T group, respectively, *p* > 0.05) (Figure 4D). Nevertheless, when serum C-peptide levels were related to insulin-positive area/total area in the pancreas (Insulin Ratio), we found that β-pancreatic islets were more efficient in treated animals, whose mean value was more than double in T group and reached statistical significance (7.7 ± 1.6 and 17.0 ± 9.7 in C and T group, respectively, *p* < 0.05) (Figure 4E). 

Taking into account the intensity of insulin-positive areas (intensity 1 being the lowest intensity degree and intensity 3 the highest intensity degree), we found no differences between intensity 1 areas when comparing both groups (21.0 and 14.8% in C and T group, respectively, *p* > 0.05). Intensity 2 areas were more abundant in C group (78.2 and 63.2% in C and T group, respectively, *p* < 0.05), and, conversely, intensity 3 areas were in greater proportion in T group (0.8 and 21.9% in C and T group, respectively, *p* < 0.01) (Figure 4F). Representative images are shown in Figure 4G,H. This finding indicates that treated animals were able to secrete more insulin with less pancreatic effort. 

### 3.4. Effects of pA1c on GLP-1, Leptin, Biochemical Markers in Serum and Intestinal Histology

Probiotic supplementation increased serum GLP-1 levels up to 26% (130.7 ± 27.2 and 164.9 ± 12.4 pg/mL in C and T groups, respectively, *p* < 0.01) (Figure 5A), while decreased serum concentrations of leptin up to 45% (42.8 ± 14.4 and 23.7 ± 16.5 ng/mL in C and T groups, respectively, *p* < 0.05) (Figure 5B). Data from lipid profile did not indicate significant differences between groups (*p* > 0.05) (Appendix A).

According to the serum GLP-1 levels, in the supplemented group immunohistochemical GLP-1 analysis of the small intestine revealed higher GLP-1^+^ area with respect to the total area (0.10 ± 0.02 and 0.15 ± 0.03% in C and T group; *p* < 0.01) (Figure 5C). Moreover, higher percentage of GLP-1^+^ cells, with respect to the total cells, was also observed in the T group (0.49 ± 0.02 and 0.71 ± 0.1% in C and T group; *p* < 0.01) (Figure 5D). Representative images are shown in Figure 5E.

Furthermore, histological analysis for the small intestine displayed a statistically significant increase in the number of PAS stained goblet cells in T group with respect to C group (11.5 ± 1.3 and 13.8 ± 1.2% in C and T group; *p* < 0.01) (Figure 6A,B). Analyzing the gut wall villus length, statistically significant higher lengths were found in the supplemented group (514.3 ± 35.6 and 588.4 ± 38.5 µm in C and T group; *p* < 0.01) (Figure 6C,D). 

### 3.5. Effects of pA1c on Hepatic Glucose-Regulating Enzymes and Inflammatory Markers

In the liver tissue, the expression at the mRNA level of all studied genes was found to be importantly altered by the probiotic treatment. Besides the high dispersion within groups, especially in C group, the analysis confirmed statistically significant differences between both experimental groups in all studied genes. The probiotic supplementation markedly reduced the expression of the three glucose-regulating studied enzymes, which decreased up to 89%, 86%, and 40% in G6P, PEPCK, and GCK, respectively (*p* < 0.001) (Figure 7A).

In addition, it seems that the treatment also attenuated hepatic inflammation by decreasing liver IL-1β (93%, *p* < 0.01) and IL-6 expression (78%, *p* < 0.001) (Figure 7B).

### 3.6. The Association between Variation of Glucose Control and Other Biochemical and Histological Parameters

Correlation coefficients are shown in Appendix A. Data from log-transformed FBG values positively correlated with serum leptin levels (*p* < 0.05), G6P (*p* < 0.01), and PEPCK (*p* < 0.01), while they negatively correlated with GLP-1 levels (*p* < 0.001) and GCK (*p* < 0.05). GLP-1 also inversely correlated with GP6 (*p* < 0.05). We only observed a significant correlation between C-peptide with G6P (*p* < 0.05), and between serum leptin and FBG levels (*p* < 0.05). Goblet cells negatively correlated with HOMA-IR (*p* < 0.05). Surprisingly, BW did not correlate with any studied parameter.

## 4. Discussion

In the present study, we have investigated the blood glucose-control effect of the probiotic pA1c in in vivo animal models through the evaluation of the main biochemical parameters implicated in the risk for PreD development, including those used for PreD diagnosis (FBG and GTT) [1], and others such as BW, HOMA-IR, or circulating C-peptide [2,47]. According to our hypothesis, we observed that pA1c administration improved glucose tolerance in HFD-induced diabetic mice for 12 weeks. Maintaining a normal glucose tolerance is key for preventing diabetes onset [2], and there are good prospects for controlling glycemia with probiotic interventions. To illustrate, findings from 21 RCTs on adults with PreD and T2D confirmed a significant reduction in FBG values with different combinations of *Lactobacillus*, *Bifidobacterium*, *Lactococcus*, *Streptococcus*, *Bacillus*, *Enterococcus*, *Propionibacterium*, and *Acetobacter* spp. [33], and the authors suggested that these effects could be a consequence of improved IR. With regard to animal studies, probiotic intervention with *Pediococcus* [48] and *Lactobacillus* spp. [49,50] in HFD-fed T2D murine models reported a reduction in FBG values, in harmony with our results and a previous study with a different *P. acidilactici* strain [51]. 

With the purpose to verify the improvement in insulin sensitivity induced by pA1c, we also studied GTT and HOMA-IR index as indirect methods for the estimation of insulin sensibility [31]. This information is relevant since IR is a risk factor for PreD [2] and is significantly involved in T2D pathogenesis [52]. A recent systematic review of RCTs and experimental studies with probiotics and other nutraceuticals failed to identify consistent and major improvements in IR in human patients [31]. Meanwhile, our study and other animal studies with probiotic treatments reported the alleviation in glucose tolerance and lower GTT and AUC values in T2D rodents [49,50,53,54], as well as the attenuation in insulin resistance (HOMA-IR values) in HFD-fed [49] and T2D mice [55]. Globally, the main results related to glucose metabolism suggest that pA1c relieved IR and attenuated metabolic derangements in HFD-induced diabetic mice, along with other improvements (discussed below). 

We also evaluated the impact of the pA1c probiotic supplementation on the mRNA expression of key glucose-metabolism regulating enzymes. PEPCK and G6P are enzymes linked to the hepatic gluconeogenesis pathway [56], thus their downregulation contributes to a reduced FBG [57]. On the other hand, GCK is the rate-limiting enzyme in glucose metabolism and acts as a glucose sensor, controlling glycogen synthesis and β-cell proliferation and function [58,59]. In our study, the probiotic administration reduced liver expression of PEPCK and G6P, genes which were previously found enriched under HFD conditions [49]. These findings suggest that pA1c induces transcriptional alterations that prevent the increase in gluconeogenesis in PreD and T2D and, therefore, has the potential to be a good anti-hyperglycemic agent. This is supported by our data from FBG, GTT, and HOMA-IR, as well as with previous literature [49,60,61,62,63]. Some of those authors linked the decrease in PEPCK and G6P expression to reduced gluconeogenesis and, consequently, lower FBG [61,63], and this is in line with our results, where FBG values positively correlated with G6P and PEPCK, and inversely with GCK. 

To continue, we also reported a drop in liver GCK expression, which correlates well with one previous investigation [49]. By comparison, other reports have observed the upregulation of hepatic GCK following the intake of a polysaccharide extract [64] and a fermented product [57] in diabetic and HFD-fed mice, respectively. Our finding suggests a reduced glycolysis or glucose utilization in the liver, which would lead to greater circulating glucose levels, as well as a compromised insulin secretion [58]. Controversy already exists about the role of GCK on glucose homeostasis. There are many studies indicating that greater pancreatic GCK levels involve improved β-cell function and proliferation, however, a recently published review [58] speculates that controlling GCK activation would also preserve β-cell function, probably by minimizing oxidative stress, then improving the pancreatic function and glucose tolerance. 

Another potential explanation for the glycemic effects observed in the T group could be the increased GLP-1 serum levels. This incretin was found depleted in T2D patients [65]. It is worth noting that our study and previous probiotic interventions in animal models also showed that GLP-1 secretion was stimulated [55,66,67,68,69]. GLP-1 has an important effect on glucose homeostasis as key responsible for insulin secretion following enteral nutrition, and also controls hepatic glucose synthesis [33,70,71]. This enteroendocrine peptide can also minimize food absorption by decreasing gastric emptying and gut motility [72], as well as promote glucose uptake in other tissues such as muscle and liver [29]. According to this, greater GLP-1 levels would turn into greater insulin levels, greater glucose utilization, lower FBG values, and improved glucose homeostasis, which matches perfectly with our findings (Figure 8). In our investigation, we observed a greater insulin production in pA1c-treated animals (T group), evidenced by greater serum C-peptide levels, as well as enhanced insulin signaling in pancreas tissue sections. It is possible that the greater levels of GLP-1 in T group may have help in restoring β-cell mass and activity [59,73]. We believe this outcome is advantageous since most T2D therapies must aim to attenuate IR or increase insulin levels by preserving β-cells function [74], whose function is typically altered in T2D [52]. Besides, GLP-1 levels inversely correlated with FBG levels, and data from GTT, AUC, and HOMA-IR indicated that glucose sensitivity was improved in T group. The same effect was also observed in previous studies that reported greater plasma and intestinal GLP-1 secretion, along with enhanced glucose tolerance [68] and controlled glycemia [55].

In addition to the aforementioned, the effect of pA1c probiotic supplementation on GLP-1 may be related to GM modulation, in agreement with previous experiments with probiotics [67], probiotic-derived components (postbiotics) [21], and prebiotic compounds [71,75,76]. Specifically, the rise in plasma and intestinal GLP-1 levels could have been instigated by greater amounts of short-chain fatty acids (SCFAs), as described in previous animal studies [55,68]. SCFAs are products of the intestinal microbial fermentation of indigestible dietary components [77]. Previous studies have shown that SCFAs can stimulate GLP-1 secretion in L-cells according to in vitro [78], animal [79,80], and human studies [81]. On top of that, SCFAs can impact insulin sensibility by promoting β-cell proliferation [52]. Even though we did not measure intestinal or fecal SCFA profile, it is likely that GM modulation, following pA1c treatment, may lead to an increased output of SCFAs that could directly affect GLP-1 secretion. Apart from the indirect effect on GLP-1 secretion through a SCFAs-mediated mechanism, pA1c could have also directly stimulated GLP-1 production through the release of microbial functional peptides or compounds, as previously described for a *Staphylococcus epidermis* strain in a recent investigation [82]. 

Although it is mainly synthesized and released in the intestine, experimental studies have demonstrated that GLP-1 has pleiotropic actions and exerts extra-intestinal effects. For example, it can reach the brain, affecting appetite and, consequently, food intake [83]. So, the greater serum levels of GLP-1 in T group could have induced lower food intake, provoking a lower BW in this group as compared with C group. SCFAs may be involved in pA1c’s effect on BW as well, since they regulate energy metabolism and energy store, and this, to some extent, could be due to their effect on incretin hormones such as GLP-1 [77]. It is well known that being overweight brings added risks for the progression of PreD to T2D. Thus, weight loss is the cornerstone in prediabetes in order to prevent T2D development [84]. While some previous human [85,86] and animal [87,88] studies with probiotic administration found no significant effect on BW, others reported a BW normalization [49,54,55] or delay in BW gain [53] in HFD-induced diabetic mice. Our finding fits well with the trend for lower BW found in pA1c-supplemented animals during the study (Figure 8). Thus, pA1c supplementation could be a good strategy to keep extra weight off and stay healthy. Previous studies have reported significant decreases in body fat percentage in PreD subjects undergoing a probiotic treatment [89] Nevertheless, further information of the body composition would confirm such an effect for our probiotic strain. There is considerable controversy about the effects of probiotics on lipid profile in T2D context, and studies provided conflicting findings [6,33]. 

In parallel, we found a marked reduction in serum leptin concentrations. Leptin is an adipokine mainly secreted by adipose cells, which exerts pleiotropic effects and is involved in many systemic functions, including metabolism and energy expenditure [90]. It is related to obesity and IR, and hyperleptinemia is frequent in HFD-fed animal models [53,91]. In preclinical trials, pA1c and other probiotic supplementations [53,92,93] significantly reduced circulating leptin levels, while other studies did not report significant changes [87,94], and evidence from human studies is weak [95]. Previously, in vitro experiments in human and animal cell lines showed that leptin induces GLP-1 production [96], which could help explain our findings. Besides the above-mentioned hypothesis, it must be taken into account that GLP-1’s half-life is considerably short [59,70], and it is likely that it is not the only contributing factor in the metabolic improvements observed following pA1c supplementation.

Another reason for the preserved pancreatic function, observed after pA1c supplementation, could be related to a best maintained intestinal barrier integrity secondary to GM modulation [33]. Our study revealed that pA1c increases the number of goblet cells, and in a similar way, a previous paper reported the normalization of the intestinal histology and preservation of goblet cells mass and mucus layer integrity in T2D rats treated with two *Lactobacillus* strains [50]. This finding is interesting because this cell type is responsible for the synthesis and secretion of mucin glycoproteins [97,98] and constitutes, along with other cell types, the main cellular components of the innate immune system [99]. On top of that, mucus layer is one of the major mediators implicated in controlling intestinal permeability and health [100,101,102]. Besides the mucus layer, there are other markers of the intestinal integrity, such as the intestinal morphology. Many researchers have described that the intestinal villi length, the depth of the crypt, and the V/C ratio are important indicators to evaluate the gut barrier function [103], as well as the digestion and absorption function of the small intestine [104]. Higher villi length has been associated to improved absorption of nutrients [105], and our results revealed that pA1c maintained greater villi length, as previously reported in growing pigs supplemented with a *P. acidilactici strain* [106]. 

In this light, a potential mechanism of action of pA1c could be related to the protection of the intestinal integrity by pA1c supplementation, reducing the paracellular permeability of luminal lipopolysaccharide (LPS). LPS is an inflammatory reagent that is increased under HFD conditions [8], provoking the induction of inflammatory pathways, activating immune cells [8,50,107] and negatively altering the gut permeability [108,109,110]. Hence, an adequate permeability would limit the amount of antigens that reach the pancreatic tissue and thus reduce pancreatic damage and systemic inflammation [31,111]. Regarding the mechanistic basis behind the modulation in the barrier function following probiotic treatments, the theories are varied. While previous authors have postulated that cell-wall elements may be behind the improvements in the barrier function observed in in vitro experiments with cell lines [112], other reports pointed out to a role for secreted molecules, such as proteins [30,113] or bioactive factors secreted to the media [108]. Certainly, all of these suppositions may be right, and there may be a specie and strain specific effect, and further investigations would confirm this.

Besides the mucus layer and TJs complexes, the GM is also an important protective factor controlling the barrier function and intestinal health [109]. It is very likely that the exposition to pA1c impacted the GM, which is an interesting therapeutic target for PreD since intestinal dysbiosis is frequent in prediabetic subjects [13,114]. 

It is known that HFD induces dysbiosis, which is usually dominated by Gram-negative and, by means of local inflammation, can lead to greater permeability [109], and increase circulating levels of endotoxin LPS [8], which, in turn, contributes to systemic inflammation and immune activation [8,50,107]. In this light, a potential hypothesis is that changes in the GM composition by pA1c treatment restored the HFD-induced dysbiosis and shifted the Gram-negative to Gram-positive ratio in T group, thus reducing the proportion of Gram-negative taxa and, consequently, decreasing LPS levels. Possible explanations are that GM modulation by pA1c treatment could have increased the proportion or activity of microbial taxa with anti-inflammatory effects [9] or changed SCFAs production, which are produced by a selected group of bacteria and can improve intestinal permeability, enhance mucus production, or control inflammation [115], as well as alter GM composition, especially in a context of HFD [116]. One study showed that *P. acidilactici* UL5 did not influence GM composition, but it did affect its metabolic activity by inducing the production of certain SCFAs [117]. In addition, previous in vitro experiments, with other *P. acidilactici* strains, showed good probiotic properties, [34,35,36,37,38,39,40], and all of these make it highly likely that exposition to PA provoked marked changes in the GM structure or activity. An important limitation of our study is the lack of a detailed analysis of the GM comparing the group treated with pA1c vs. the control group. The data obtained from this analysis would have helped us to better understand the mechanism of action of pA1c in the animal model used. We hope to be able to address this problem in future experiments.

However, although our results show a surprising benefit of the administration of the pA1c, some limitations have to be considered. Regarding the histological and functional analysis of pancreas, β-cell mass regulation considerably differs between mouse models and humans [74], and we must be cautious and sensible when extrapolating findings from animal studies. Additional studies must also determine the counts of pA1c on the GM to verify its impact on the GM, as well as to clarify the mechanism of action by which this strain improves glucose homeostasis and attenuates the diabetic phenotype in HFD-induced diabetic mice. In spite of this, we consider our findings to have given us a solid picture of the beneficial effects provided by pA1c in the studied model (Figure 8), which has bright future perspectives. 

We reckon that pA1c could be combined with other probiotic strains or prebiotics to create a cocktail of bioactive components, in a similar manner as described in many human studies [29,33]. The combination of different bacteria genera or species may enhance the observed beneficial effects or produced additional positive effects. On another note, it would be interesting to test whether probiotic viability is required to maintain pA1c’s health-promoting properties. Inactivated microorganisms, i.e., pasteurized [118], irradiated [119], or probiotic-derived products, often referred to as postbiotics [120,121], have been demonstrated to confer many benefits to the host in T2D [21], as well as in different contexts (reviewed in [122]), and may present a potent strategy for the control of metabolic diseases.

## 5. Conclusions

Altogether, the results presented in this study have demonstrated that pA1c counteracted BW gain and insulin resistance, in a model of HFD-induced diabetic mice, thus preserving the intestinal function and metabolic health. The mechanisms that explain the antidiabetic activity of pA1c remain partially unexplored; however, we speculate that they may include protection against pancreatic degeneration and alteration of the intestinal barrier dysfunction, followed by metabolic improvements and GM alterations. Notwithstanding the foregoing, more experiments are needed to clarify the mechanisms by which pA1c improves glucose homeostasis and attenuates the diabetic phenotype in this mouse model. Besides the obvious limitations of animal studies, they still constitute an important source of information and contribute mechanistic insights into the effects of probiotics on PreD and T2D progression. Microbiotherapy has become a hot topic among researchers, clinicians, and biotechnology companies, and it holds promise as strategy for controlling the diabetes pandemic. However, the dietary strategy of probiotic supplementation with pA1c, for improving glucose control and preventing metabolic disorders in PreD and T2D, requires further investigation.

It is also important to mention that this was a preliminary, proof-of-concept investigation, and the application of pA1c in a future clinical trial would help us to elucidate whether the benefits observed in this animal model are transferable to pre-diabetic and diabetic patients. In this sense, the improvement of the health variables “fasting glycemia” and “glycosylated hemoglobin”, among others, should be the main variables to observe after the administration of the *Pediococcus acidilactici* pA1c in these clinical trials.

## 6. Patents

Part of the results that serve as the basis for this manuscript have been included in the patent entitled: *Probiotics for regulating blood glucose* [PCT/EP2020/087284; WO2021123355A1].

## Figures and Tables

**Figure 1 nutrients-14-00692-f001:**
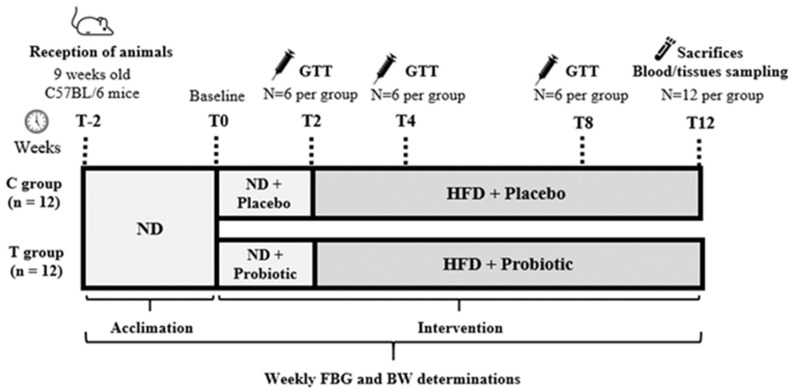
Experimental design. BW: body weight; FBG: fasting blood glucose; GTT: glucose tolerance test; HFD: high-fat diet; ND: normal diet.

**Figure 2 nutrients-14-00692-f002:**
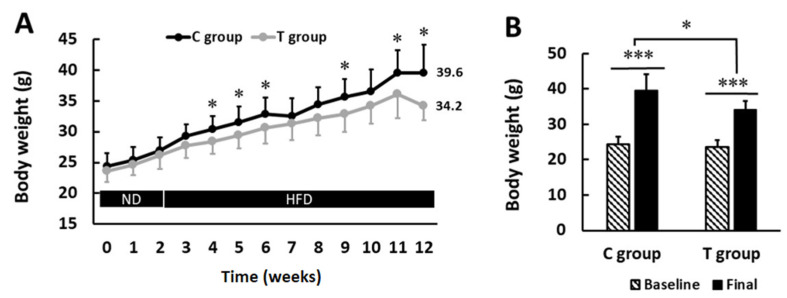
Body weight (BW) progression and BW gain during the intervention. (**A**) Statistically significant differences were found between experimental groups at 4, 5, 6, 9, 11, and 12 weeks (*p* < 0.05). (**B**) BW gain was statistically significant in both C group and T group (*p* < 0.05). Values are expressed as mean ± SD. HFD: high-fat diet; ND: normal diet. * *p* < 0.05, *** *p* < 0.001.

**Figure 3 nutrients-14-00692-f003:**
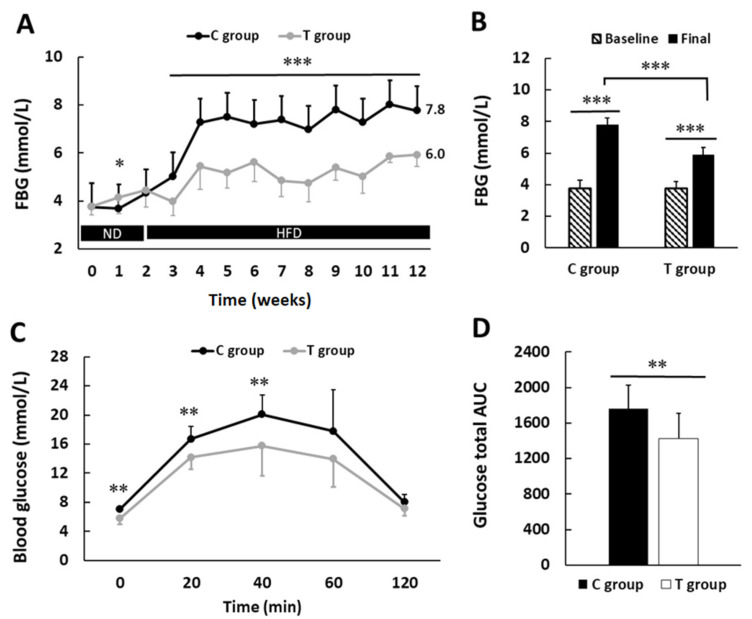
Effect of probiotic supplementation on glucose metabolism. (**A**) Weekly measured fasting blood glucose (FBG) mean values. (**B**) Bar plots represent basal and final FBG values in both groups. (**C**) 8 weeks glucose tolerance test (GTT) curve. (**D**) Area under the curve (AUC) plot. HFD: high-fat diet; ND: normal diet. Values are expressed as mean ± SD. * *p* < 0.05, ** *p* < 0.01, *** *p* < 0.001.

**Figure 4 nutrients-14-00692-f004:**
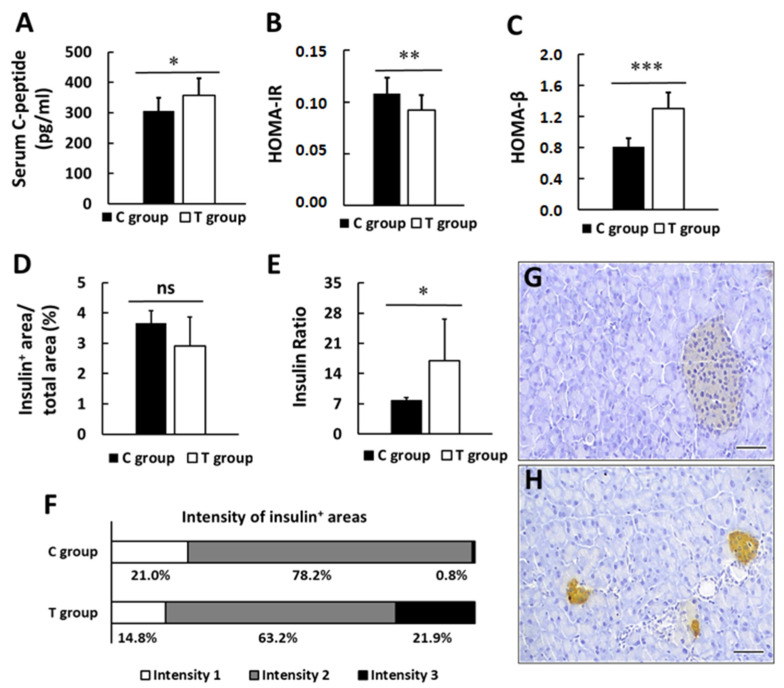
Impact of probiotic supplementation on pancreatic activity. (**A**) Serum C-peptide levels. (**B**) HOMA-IR index. (**C**) HOMA-β index. (**D**) Insulin positive area per total area. (**E**) Insulin Ratio (C-peptide level per insulin positive area/total area). (**F**) Intensity of insulin positive areas. Representative images of pancreas sections immunohistochemically-stained for insulin C group (**G**) and T group (**H**). Values are expressed as mean ± SD. * *p* < 0.05, ** *p* < 0.01, *** *p* < 0.001. ns: no significance. Scale bars: 50 µm.

**Figure 5 nutrients-14-00692-f005:**
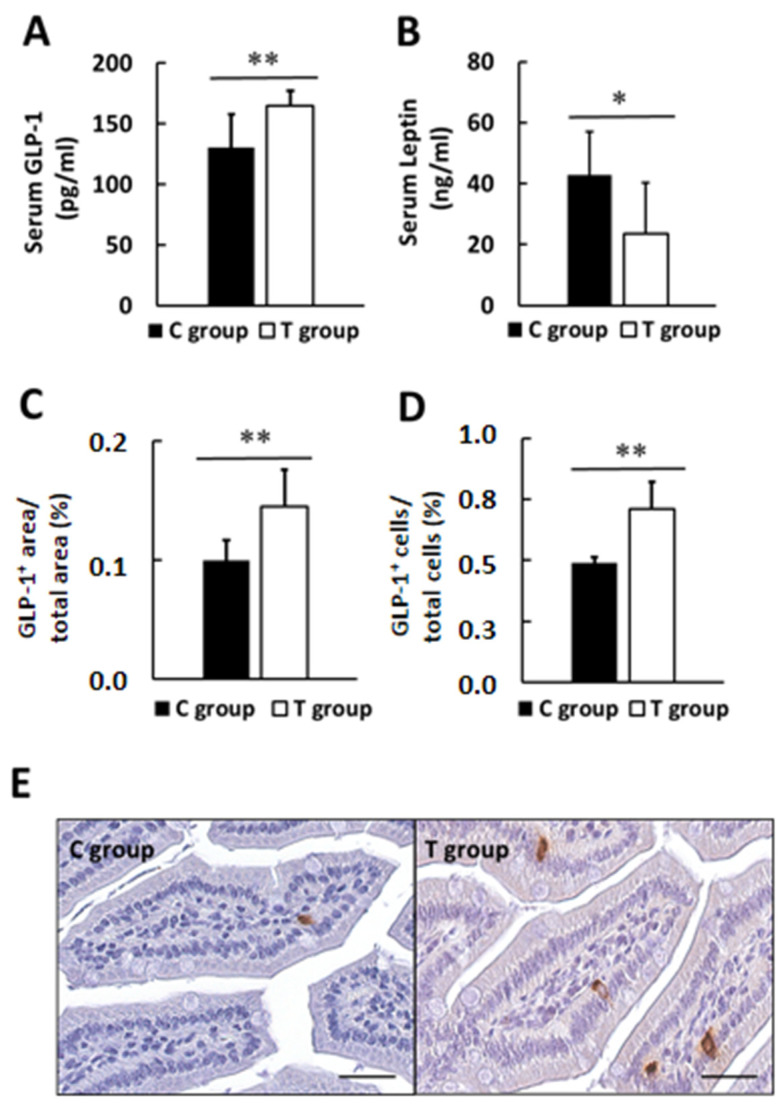
(**A**) Effect of probiotic supplementation on serum GLP-1 levels and (**B**) leptin levels. (**C**) GLP-1 positive area (%) and (**D**) GLP-1 positive cells (%) in intestinal histological samples. (**E**) Representative immunostaining for GLP-1 in small intestine. Values are expressed as mean ± SD. * *p* < 0.05, ** *p* < 0.01. Scale bars: 25 µm.

**Figure 6 nutrients-14-00692-f006:**
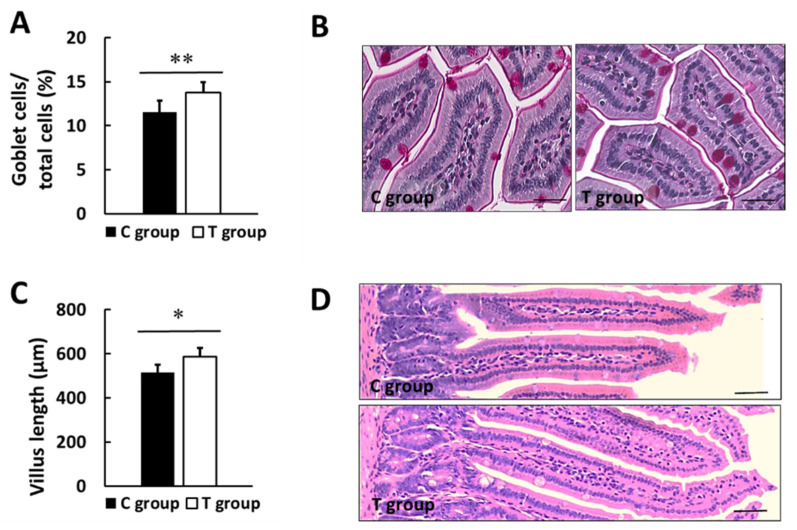
Intestinal histology. (**A**) Goblet cells percentage (%). (**B**) Representative images of PAS stained small intestine sections. (**C**) Length of gut wall villus. (**D**) Representative images of H&E-stained gut villus. Values are expressed as mean ± SD. * *p* < 0.05, ** *p* < 0.01. Scale bars: 25 µm (**B**) and 50 µm (**D**).

**Figure 7 nutrients-14-00692-f007:**
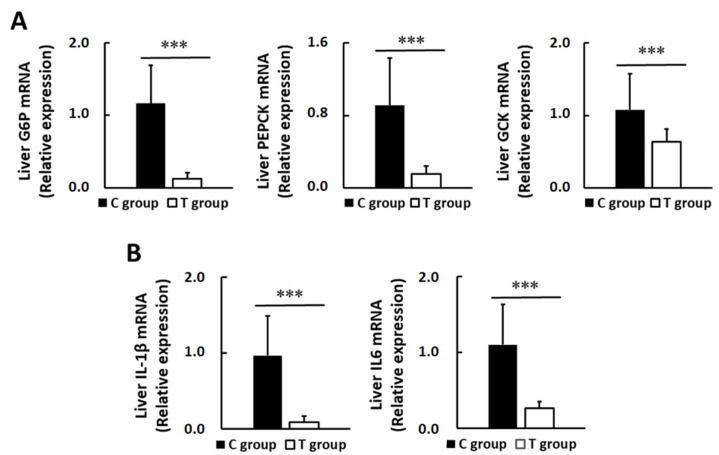
Hepatic gene expression. (**A**) Transcriptional levels of hepatic glucose regulating enzymes and (**B**) inflammatory markers. Values are expressed as mean ± SD. *** *p* < 0.001.

**Figure 8 nutrients-14-00692-f008:**
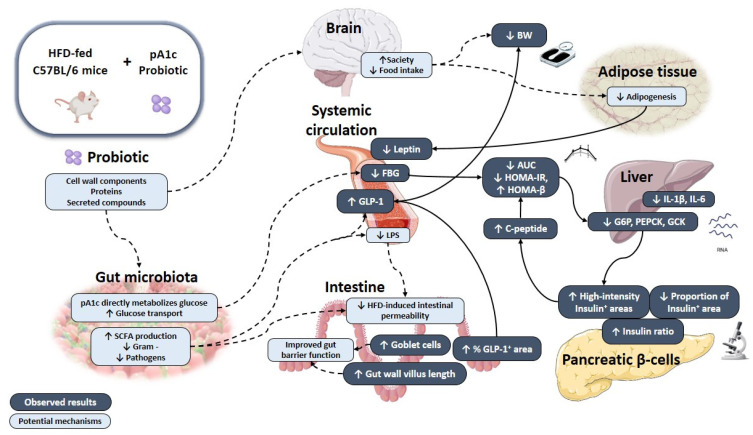
Schematic representation of the mechanism of action through which pA1c exerts its beneficial effects on HFD-induced mice. The graphical abstract includes pathways demonstrated with results observed in this work (black line), as well as potential mechanisms described by other studies (dotted line). Up arrow: increased expression; Down arrow: decreased expression.

## Data Availability

Data is contained within the article or supplementary material.

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
