# Peer review of "Antidiabetic Effects of Pediococcus acidilactici pA1c on HFD-Induced Mice"

_nutrients, 2022, doi:10.3390/nu14030692_

Round 1
Reviewer 1 Report
This is a well written paper with appropriate methods and statistical analysis. The authors demonstrate in a mouse model that pA1c attenuated body weight gain and insulin resistance. My only comment is to add a sentence in the discussion to discuss the potential implications and impact for clinical practice if this work is extended beyond the mouse model to humans and proves effective.
Author Response
January 19th, 2022
Journal: Nutrients (ISSN 2072-6643)
Title: Antidiabetic effects of Pediococcus acidilactici pA1c on HFD-induced mice
Authors: Miriam Cabello-Olmo, María Oneca, María José Pajares, Maddalen Jiménez, Josune Ayo, Ignacio J. Encio, Miguel Barajas* and Miriam Araña*
Manuscript ID: nutrients-1544857
REVIEWER #1
This is a well written paper with appropriate methods and statistical analysis. The authors demonstrate in a mouse model that pA1c attenuated body weight gain and insulin resistance. My only comment is to add a sentence in the discussion to discuss the potential implications and impact for clinical practice if this work is extended beyond the mouse model to humans and proves effective.
We very much appreciate the prompt consideration of our manuscript and constructive criticism of the reviewer. Detailed below is a summary of the changes that have been made in the manuscript with appropriate rebuttal comments. The changes made on the manuscript addressing point-by-point according to the reviewer comments are itemized below.
We also appreciate the critical spirit of the review and the words of support for the manuscript regarding its writing, methodology and statistical analysis.
Regarding the reviewer´s comment to add a sentence in order to discuss the potential implications and impact for clinical practice if this work is extended beyond the mouse model to humans and proves its effectiveness, it is worth noting that we already indicated at the end of the Conclusion section that “…the application of pA1c in a future clinical trial would help us to elucidate whether the benefits observed in this animal model is transferable to pre-diabetic and diabetic patients”. However, in order to emphasize this sentence and responding the reviewer, we have added the following sentence at the end of the Conclusion section: “In this sense, the improvement of the health variables “fasting glycemia” and “glycosylated hemoglobin”, among others, should be the main variables to observe after the administration of the Pediococcus acidilactici pA1c in these clinical trials”.
We hope that this improved manuscript satisfy the requests and is acceptable for publication in Nutrients.
Reviewer 2 Report
This paper is focused on antidiabetic effects of Pediococcus acidilactici pA1c in a murine model of high-fat diet (HFD)-induced T2D. This study is very well written. The figures and the experimental design are suitable and they are very useful for understanding the study. Only minor revisions are needed to clarify some points of the study:
- Was microbiota analysis performed in controls and probiotic treated mice?
- In figure 8 change “is beneficial” lane 548 with “its beneficial”
Author Response
January 19th, 2022
Journal: Nutrients (ISSN 2072-6643)
Title: Antidiabetic effects of Pediococcus acidilactici pA1c on HFD-induced mice
Authors: Miriam Cabello-Olmo, María Oneca, María José Pajares, Maddalen Jiménez, Josune Ayo, Ignacio J. Encio, Miguel Barajas* and Miriam Araña*
Manuscript ID: nutrients-1544857
REVIEWER #2
This paper is focused on antidiabetic effects of Pediococcus acidilactici pA1c in a murine model of high-fat diet (HFD)-induced T2D. This study is very well written. The figures and the experimental design are suitable and they are very useful for understanding the study. Only minor revisions are needed to clarify some points of the study:
We very much appreciate the prompt consideration of our manuscript and constructive criticism of the reviewer. Detailed below is a summary of the changes that have been made in the manuscript with appropriate rebuttal comments. The changes made on the manuscript addressing point-by-point according to the reviewer comments are itemized below.
Was microbiota analysis performed in controls and probiotic treated mice?
The question asked by the reviewer regarding the analysis of the microbiota is a key point in all studies in which an attempt is made to influence the intestinal microbiota with the aim of obtaining positive results in some health variable, as in our case insulin resistance and overweight. Without any doubt, we are very interested in analyzing this variable in order to find an answer to the complex mechanism of action that could be responsible for the beneficial effect observed after the administration of Pediococcus acidilactici pA1c. We have started the analysis of the microbiota comparing the control group vs. the treated group but unfortunately we have not yet received a response from the bioinformatic service. The only data we have to date in this regard are:
- We have been able to detect the presence of Pediococcus acidilactici (by RTqPCR) in stool samples obtained from mice treated with Pediococcus acidilactici pA1c and, obviously, not in samples from control mice. This clearly indicates that the microorganism supplemented in the food is able to pass through the digestive tract and survive.
- Our preliminary results indicate that there are relevant changes in the diversity and abundance of bacterial genera in stool samples from mice treated with Pediococcus acidilactici pA1c vs samples obtained from control mice. These data are being carefully analyzed and we hope to have results in the coming weeks.
In figure 8 change “is beneficial” lane 548 with “its beneficial”
Thanks for your appreciation, we have modified this item in the new version of the manuscript.